# Cranial functional specialisation for strength precedes morphological evolution in Oviraptorosauria
Luke E. Meade [1] ✉, Michael Pittman [2], Amy Balanoff[3] & Stephan Lautenschlager [1]

Oviraptorosaurians were a theropod dinosaur group that reached high diversity in the Late Cretaceous. Within oviraptorosaurians, the later diverging oviraptorids evolved distinctive crania which were extensively pneumatised, short and tall, and had a robust toothless beak, interpreted as providing a powerful bite for their herbivorous to omnivorous diet. The present study explores the ability of oviraptorid crania to resist large mechanical stresses compared with other theropods and where this adaptation originated within oviraptorosaurians. Digital 3D cranial models were constructed for the earliest diverging oviraptorosaurian, *Incisivosaurus gauthieri*, and three oviraptorids, *Citipati osmolskae*, *Conchoraptor gracilis*, and *Khaan mckennai*. Finite element analyses indicate oviraptorosaurian crania were stronger than those of other herbivorous theropods (*Erlikosaurus* and *Ornithomimus*) and were more comparable to the large, carnivorous *Allosaurus*. The cranial biomechanics of *Incisivosaurus* align with oviraptorids, indicating an early establishment of distinctive strengthened cranial biomechanics in Oviraptorosauria, even before the highly modified oviraptorid cranial morphology. Bite modelling, using estimated muscle forces, suggests oviraptorid crania may have functioned closer to structural safety limits. Low mechanical stresses around the beaks of oviraptorids suggest a convergently evolved, functionally distinct rhamphotheca, serving as a cropping/feeding tool rather than for stress reduction, when compared with other herbivorous theropods.

Oviraptorosauria, a clade of maniraptoran theropod dinosaurs from the Cretaceous of Laurasia[1,2], is currently represented by approximately 50 genera[2,3], which range from chicken-sized forms to the approximately eight-metre long, two-tonne *Gigantoraptor erlianensis*[4]. The group has become famous for their preserved eggs and nesting behaviours, from which circumstances they also derive their name[5–8]. The superficially bird-like cranium of oviraptorosaurians, and Oviraptoridae in particular, has a unique morphology among Dinosauria and is the most aberrant part of their skeleton[1,9–11]. Their shortened toothless rostrum and palate is dominated by a robust premaxilla and their crania have expanded space for jaw adductor musculature, which suggest adaptation towards high bite forces[12–16]. This inference is supported by bite force estimates from digital volumetric reconstruction[17]. However, expanded pneumatic spaces[18–20], a delicate often ornamented skull roof[1,3,21–23], thin rod-like jugals and quadratojugals[18,24,25], and large orbits[19] make the cranium lightweight but could potentially reduce

cranial strength by creating hotspots of mechanical stress during the contraction of powerful jaw adductor musculature and associated biting reaction forces.

The presence of powerful jaws in oviraptorids has been used to suggest specialist crushing diets (e.g., molluscivory[6]), but most interpretations focus on a herbivorous diet[26]. The powerful bite of oviraptorids subsequently is often contextualised as an adaptation to procure and process a wide range of foodstuffs such as seeds, nuts, fruits, stems, and bark[15,27–30]—similar to modern parrots[31]—or to shear xerophytic vegetation[32]. This ecology contrasts with other herbivorous theropod dinosaurs, which are reconstructed with weak bite forces (e.g., the therizinosaurian *Erlikosaurus*[33] and ornithomimosaurians[34]).

The question remains how the highly modified oviraptorid cranial architecture is adapted to resist the mechanical stresses associated with powerful jaw muscles, in comparison to other theropod groups and whether

[1]School of Geography, Earth and Environmental Sciences, University of Birmingham, Birmingham, UK. [2]School of Life Sciences, The Chinese University of Hong Kong, Hong Kong SAR, China. [3]Center for Functional Anatomy and Evolution, Johns Hopkins University School of Medicine, Baltimore, USA. ✉e-mail: luke.edward.meade@gmail.com

this was established early in the oviraptorosaurian lineage. Finite element analysis (FEA) has been used to assess the performance of structures in a wide variety of tetrapod groups[35] and is a useful method for comparing functional morphology, particularly in investigating and comparing feeding in extinct herbivorous non-avian dinosaur groups[36–41]) and other herbivorous archosaurs[42]. Here, we use FEA to quantify cranial performance in the oviraptorids *Citipati osmolskae*[43], *Khaan mckennai*[43], and *Conchoraptor gracilis*[13] from the Late Cretaceous of Mongolia, along with the earliest-diverging oviraptorosaurian, *Incisivosaurus gauthieri*[44] from the Early Cretaceous of what is now northern China.

*Incisivosaurus* has a cranial morphology that combines features of non-oviraptorosaurian tetanuran theropods and the more unorthodox oviraptorids[44,45]. *Incisivosaurus* retains teeth, which are unusual and heterodont[44], and has a more typical theropod jugal shape[44–46], not the rod-like form of the oviraptorids[18,24]. *Incisivosaurus* has some degree of cranial shortening and a relatively large circular orbit. The cranium, however, is not dorsoventrally expanded, and the bones of the skull roof are less pneumatised than in Oviraptoridae[19,45].

First, we directly investigate cranial strength using scaled loading, a non-physiological strength test independent of muscular and behavioural differences, to directly compare the stress resistance of different cranial shapes. Based on the previous suggestion that the unusual oviraptorid cranial form is adapted for high bite forces[12–15], we would expect its structure to resist the mechanical stresses associated with the reaction force from biting comparatively well. We test the hypothesis that oviraptorid cranial morphology, adapted as part of a system to produce powerful sustained bite forces, will be better able to resist bending forces and experience lower mechanical stresses compared with the earlier-diverging oviraptorosaurian *Incisivosaurus* and other non-oviraptorosaurian theropods. As a subhypothesis, we expect results for the tooth-bearing, less modified theropod cranial shape of *Incisivosaurus* to represent an intermediate stage or be more like those of the non-oviraptorosaurian theropods.

Secondly, we test the hypothesis that the oviraptorid cranium will experience similar stress magnitudes as other theropod taxa undergoing muscle-driven bites, resisting the stresses produced by comparatively strengthened adductor musculature[17] when all models are loaded using taxon-specific estimated muscle force vectors. If our estimates of muscle force are accurate[17], this is to be expected because of the conservation of safety factors influencing skeletal functional morphology[39,47]. We use FEA scenarios based on volumetric estimates of jaw adductor muscle force in which differing performance between bite positions may give clues towards comparative functional ecology in Oviraptorosauria.

## Methods
3D meshes of retrodeformed oviraptorosaurian crania were generated using Avizo Lite (version 9.3.0) and Blender (version 2.9.0)[48] based on CT scans of the crania of *Incisivosaurus* (IVPP V13326), *Citipati* (MPC-D 100/798), and *Khaan* (MPC-D 100/973). The cranial model for *Conchoraptor* is a composite derived from a CT scan of MPC-D 100/3006 and photogrammetric data for the palate and premaxilla of ZPAL Mg-D I/95; all extensively pneumatised elements and the braincase were derived from CT scan data as not capturing this internal anatomy may affect FEA results[49]. Full details of scan datasets and retrodeformational procedure can be found in the supplement of ref. 17 and figured in ref. 50. Our oviraptorosaurian cranial models do not include sutural connections as many were unclear in the CT datasets and could not be consistently reconstructed across all taxa, either being matrix-bound areas with low contrast or so damaged as to be unrecognisable in large regions. Furthermore, it allowed for comparison with key previously published non-oviraptorosaurian FE models that did not model cranial sutures[40,42,51]. Using the Avizo simplification editor, cranial models were simplified to <400,000 triangular faces, enough to provide a density of tetrahedral elements to balance sufficiently detailed FEA results[52,53] with easier data processing and analytical speed. Cranial meshes were checked to be manifold and cleaned in the Avizo mesh editor. The

cranial surface meshes were converted to a structure of four-noded tetrahedral elements (tet4) in Hypermesh (version 13.0.110).

The four oviraptorosaurians are compared with *Erlikosaurus andrewsi*[54] and *Ornithomimus edmontonicus*[55], which represent theropod groups—Therizinosauria and Ornithomimosauria—that are adapted towards herbivory[30,56] and are partially (*Erlikosaurus*) or fully (*Ornithomimus*) edentulous. The cranium of *Allosaurus fragilis*[57], a large carnivorous theropod, is also used for comparison as it has a different approach to cranial strength adaptation, as part of a diet focussing on large prey, and has been interpreted previously as overengineered based on FEA[58].

Cranial models of *Allosaurus* and *Erlikosaurus* were sourced from previously published FEA studies[33,40,51,59]. The *Ornithomimus* model was supplied by A. Cuff and was used previously in published FEA and muscle reconstruction studies[34,42]. Experimentally derived material properties were assigned in Hypermesh to bone and teeth (for dentulous crania) components of each model based on extant alligator mandibles (E = 20.49 GPa, $\upsilon = 0.40$)[60,61] and extant crocodile teeth (E = 60.40 GPa, $\upsilon = 0.31$)[60,62,63]. These were considered isotropic and homogenous[64]. Model loads and constraints were set up in Hypermesh (detailed below) and the FEA was solved and visualised in Abaqus (version 6.14).

For the comparative bending test, extrinsic loads were applied perpendicular to the palate. Loads were scaled with cranial surface area (stress scales with area)[65], up from an arbitrary but realistic force of 100 N on the smallest cranium, *Incisivosaurus*, ensuring all models experienced the same relative load (summarised in Supplementary Table S1). This removed the effect of size in comparing cranial strength and efficiency and made this test independent of differences in the force vectors and proportional strengths of jaw adductor muscles, focussing purely on how well different cranial morphologies function in response to bending. In each scenario, loads were applied bilaterally to the anterior tip of the beak (or front teeth), the lateral edge of the beak midpoint (or middle of tooth row), and the posterior extent of the palate (the distinctive tooth-like projection of the maxillae and vomer in the oviraptorids and the rear teeth in the others). We also assessed the effect of applying the load unilaterally (left side). Cranial models were constrained in all axes at four points on the articular surface of each quadrate (representing jaw joint stability during biting) and three points on the occipital condyle (representing bony/muscular support of the cranium).

For the intrinsic muscle force biting test, force vectors for the oviraptorosaurians were based on Meade and Ma[17], divided across eight nodes on their origin site totalling the estimated contraction force of each jaw adductor muscle on each side. Four nodes on each quadrate articular surface and three nodes on the posterior of the occipital condyle were constrained in all directions. One or two nodes (unilateral and bilateral biting respectively) were constrained on the anterior, middle, or posterior part of the palate in the vertical (Z) axis (at the same locations as in the comparative bending test). Force vectors for *Allosaurus*, *Erlikosaurus*, and *Ornithomimus* were set up similarly based on previous FEA studies[42,51,60].

A number of further scenarios using extrinsic loading were created to model a head-pull, head-shake, and a head-twist motion (similar to refs. 51,60,66) to test the performance of the crania in ways that may be relevant to feeding, driven by postcranial musculature. In addition, cranial models assessing the influence of a keratinous rhamphotheca covering the beak of the oviraptorids on the distribution and magnitude of stress in their crania (similarly to ref. 60) were also tested. The method and results of these are described in Supplementary Text S1.

Results from FEA are compared qualitatively through visual comparison of contour plots of von Mises stress and quantitatively by mean and peak stress values and total strain energy[65]. Artificially high stress values can result as artefacts from small element size and using point loads and constraints[65,67]. To account for this and make peak stress a useful metric, the top 5% of values were excluded when comparing stress values (and calculating means) from exported reports of the von Mises stress at every element node for each FEA scenario, a solution like other FEA studies (see refs. 51,67–69). An additional correction factor based on equation (5) from Dumont et al.[65] was applied to values of total strain energy from the FEA

models using surface area scaled loads for them to be comparable, as strain energy scales with volume, not area. Contour plots showing stress distribution are figured using the Viridis colour scheme to enhance objective interpretation and accessibility[70].

## Reporting summary
Further information on research design is available in the Nature Portfolio Reporting Summary linked to this article.

## Results
Due to the minimal differences between bi- and unilateral loading in the anterior and posterior positions, the mid-palate position is the only unilateral scenario figured here; the others are figured in Supplementary Figs. S1 and S2, along with a table of mean and peak stress values for all scenarios in Supplementary Tables S2 and S3.

### Comparative bending test using scaled loading
The four oviraptorosaurian crania display lower stress magnitudes than the herbivorous theropods *Erlikosaurus* and *Ornithomimus*, with the greatest difference under anterior loading. Of the oviraptorosaurians, stress is consistently lowest in *Conchoraptor*, followed by *Khaan* (Figs. 1a, b and 2). Mean and peak stress are slightly higher in *Citipati* and *Incisivosaurus* compared with the other two oviraptorosaurians, but which taxon is greater depends on loading position (Figs. 1c, d and 2). *Allosaurus* experiences similar mean stress to the oviraptorids, between the values of *Conchoraptor* and *Khaan* under anterior and mid-palate loading, but experiences the lowest mean von Mises stress of all species under posterior loading (Figs. 1e

and 2). Mean and peak stress are consistently highest and very similar in *Erlikosaurus* and *Ornithomimus* (Figs. 1f, g and 2).

Under mid-palate loading, *Citipati* experiences both the highest mean and peak stress of any oviraptorosaurian (higher than the early diverging *Incisivosaurus*) (Figs. 1c and 2). When the mid-palate of *Citipati* is loaded unilaterally, it uniquely experiences higher mean stress than under the anterior scenario (Figs. 1c and 2). Posteriorly, stress is also slightly increased by loading unilaterally (Fig. 2), though this is only noticeable in crania in which the loads are posteriorly applied to both or one side of a tooth row rather than a single tooth-like projection in the back of the palate (the three oviraptorids).

The oviraptorosaurian crania are characterised by patterns of highest stress around the quadrate's anterior contact with the pterygoid (Fig. 1a–d). The epipterygoid and parasphenoid rostrum in all four oviraptorosaurians also experience high stresses (Fig. 1a–d), but these bones are thin and may have been poorly ossified or more flexibly connected to the rest of the cranium than is modelled. A smaller stress hotspot exists on the pterygoid just anterior to its contact with the epipterygoid. Despite the thin rod-like shape of the oviraptorid jugal and quadratojugal, this structure does not experience broadly high stress, only localised hotspots at its anterior and posterior ends (Fig. 1a–c). Furthermore, the oviraptorosaurians generally experience very low stress in the premaxilla, maxilla, and in the nasal and cranial roof (the most pneumatised areas) (Fig. 1a–d). These low-stress regions are likely to have been covered by a keratinous rhamphotheca in life, suggesting broad stress reduction was not an important function of the oviraptorid beak covering. Modelling a keratinous covering on the beak of the oviraptorids in FEA scenarios provides only minor stress reduction

**Fig. 1 | Von Mises stress distribution from comparative bending test using scaled loading.** Von Mises stress (MPa) contour plots from FEA modelling a comparative bending test on cranial models of oviraptorid oviraptorosaurians *Conchoraptor* (**a**), *Khaan* (**b**), *Citipati* (**c**), early diverging oviraptorosaurian *Incisivosaurus* (**d**), and non-oviraptorosaurian theropods *Allosaurus* (**e**), *Erlikosaurus* (**f**), and *Ornithomimus* (**g**). Results from loads applied bilaterally for three different locations on the palate and unilaterally at the mid-palate. Applied forces scaled so ratio of cranial surface area:force applied was identical in all. Scale bars on the right are 50 mm.

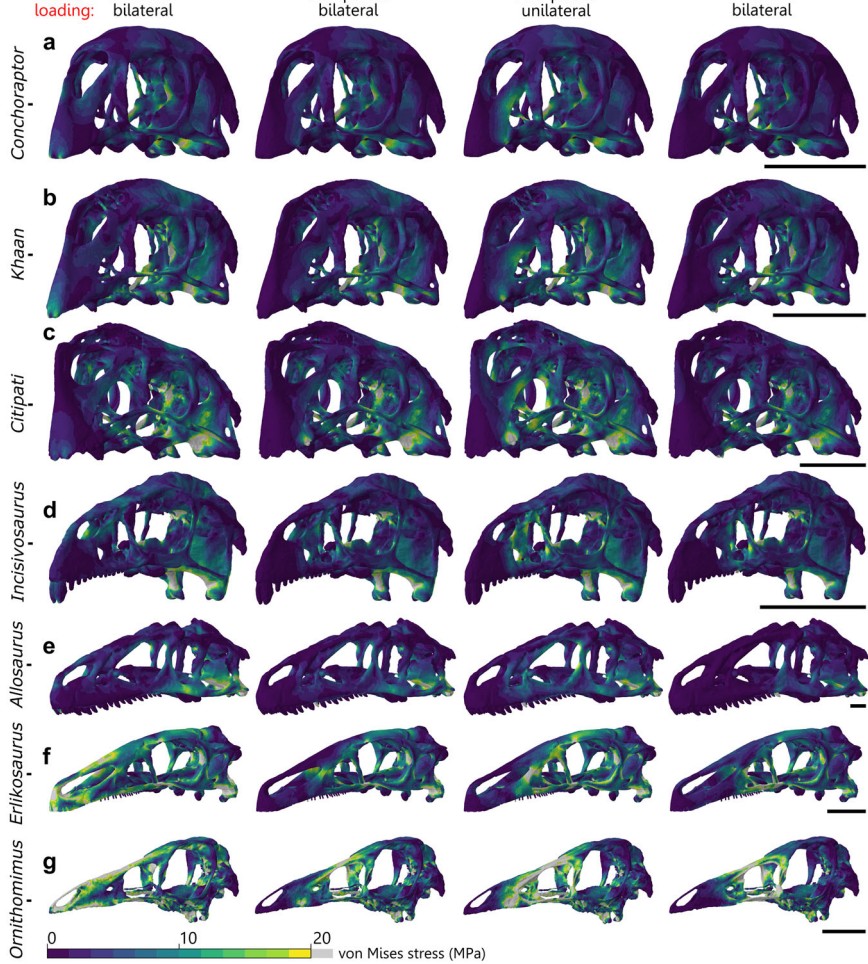

**Fig. 2 | Von Mises stress data from comparative bending test using scaled loading.** Mean values and whiskers showing first quartile, median, and third quartile of von Mises stress (MPa) from FEA of bilaterally and unilaterally applied palatal bending forces (scaled to relative surface area) at three positions on the palate of cranial models of oviraptorosaurians *Conchoraptor*, *Khaan*, *Citipati*, and *Incisivosaurus*, along with *Allosaurus*, *Erlikosaurus*, and *Ornithomimus*. Ant = anterior, Mid = mid-palate, Pos = posterior.

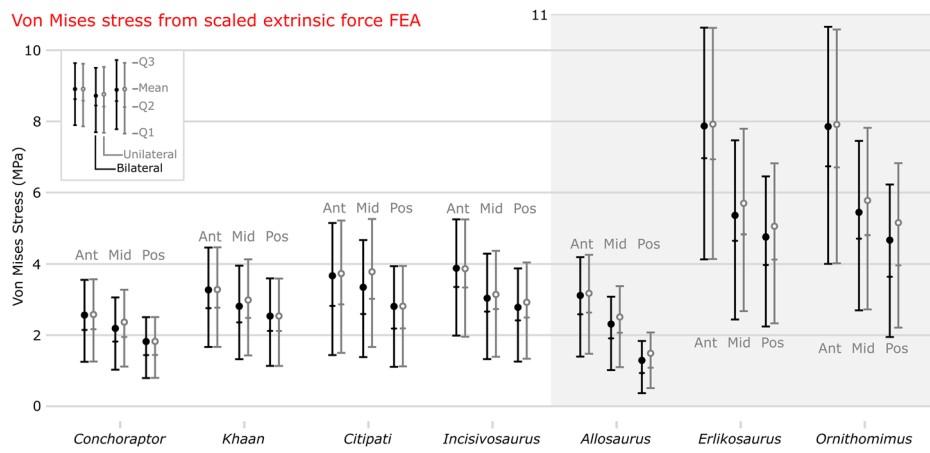

under its immediate surface, not substantially affecting the rest of the cranium (Supplementary Figs. S5 and S6).

The non-oviraptorosaurian theropods also experience stress hotspots in the quadrate and pterygoid, but these stresses are greater in *Erlikosaurus* and *Ornithomimus* relative to the oviraptorosaurians (Fig. 1f, g). In addition, the quadratojugal and jugal along with the cranial roof, lacrimal, and postorbital also experience relatively higher stress (Fig. 1f, g). Under anterior loading, the premaxilla, maxilla, and nasal are also more stressed; this is especially the case in *Erlikosaurus* and *Ornithomimus* (Fig. 1f, g). These stresses in the nasal (and maxilla) are especially severe in *Ornithomimus* for all loading positions (Fig. 1f, g). The high stresses of these regions, that would likely have been covered by a keratinous rhamphotheca in life, contrast with the low stress of their counterparts among the oviraptorids.

Though its cranial morphology has similarities to the non-oviraptorosaurian theropods, the early diverging oviraptorosaurian *Incisivosaurus* shows patterns of cranial stress more like those of the oviraptorids (Fig. 1d), implying adaptations similar to those that characterise the structural performance of oviraptorid cranial morphology are already present among early diverging oviraptorosaurians. All three non-oviraptorosaurian theropods show higher stresses on the jugal and quadratojugal for all loading positions (Fig. 1e–g), whereas *Incisivosaurus*, which has a more typical unreduced theropod jugal morphology, shows very little stress on the jugal/quadratojugal in any scenario (Fig. 1d). However, the nasals and nasal process of the premaxilla are more stressed in *Incisivosaurus* (Fig. 1d) compared with the oviraptorids (Fig. 1a–c). Under anterior loading specifically, *Incisivosaurus* shows slightly elevated stress in the nasal arch, but lower stress in the body of the premaxilla compared to the oviraptorids (Fig. 1d). This is likely linked to *Incisivosaurus* possessing teeth, which are modelled separately from bone in the FEA. Loading the prominent incisor-like teeth of *Incisivosaurus* does not produce any aberrant stress patterns (Fig. 1d), suggesting they would not be unsuitable for feeding.

Total strain energy is lower in the oviraptorosaurian crania than the other theropods under anterior loading (Supplementary Table S2), indicating a more energy efficient structure (less energy lost to bending). Under mid-palate loading, the total strain energy of *Allosaurus* falls below that of *Citipati*, which is consistently the highest value among the oviraptorosaurians. Under posterior loading, *Allosaurus* has the lowest total strain energy of any taxon, likely a result of the rear of the *Allosaurus* tooth row being relatively more posterior than in the oviraptorids, whose shortened rostrum confines the entire hard palate to a smaller but more anteriorly located region. Total strain energy in *Erlikosaurus* and *Ornithomimus* is consistently much higher than the other taxa. *Ornithomimus*, with the highest total strain energy throughout, is the only taxon to have a higher total strain energy under posterior loading compared with mid-palate loading.

### Biting test using estimated muscle force vectors

There is less difference in stress magnitude between taxa from FEA incorporating estimated muscle force vectors (rather than the comparative bending test using surface area scaled loading)—this is to be expected as a result of the conservation of safety factors influencing skeletal functional morphology[39,47].

Oviraptorids generally experience slightly greater mean and peak von Mises stress and total strain energy in all bite positions compared to the other theropods (Figs. 3 and 4). Mean stress is consistently greatest in *Citipati* and then *Khaan* (Figs. 3c, d and 4). The oviraptorid with the lowest mean stress is *Conchoraptor* (Fig. 3a), similar to that of *Erlikosaurus* (Fig. 3f) (though peak stress and total strain energy are generally higher in *Conchoraptor*); both *Conchoraptor* and *Erlikosaurus* experience greater mean von Mises stress under biting than early diverging oviraptorosaurian *Incisivosaurus* (Figs. 3d and 4). *Allosaurus* experiences the second lowest mean stress (Fig. 3e) while *Ornithomimus* is notably the least (Figs. 3g and 4), resulting from comparatively very weak musculature.

*Conchoraptor* and *Khaan* show fairly consistent stress magnitudes at all bite points (Figs. 3a, b and 4). *Allosaurus* is the only species to have notably less stress in the posterior bites compared with the other locations (Figs. 3e and 4). *Citipati* is unique in performing at its best in the anterior biting position and that the posterior bite point produced noticeably greater mean stress than the other bite points (Figs. 3c and 4).

Mean stress tends to be slightly greater in unilateral biting scenarios compared with bilateral (Fig. 4), though this is most noticeable in mid-palate bites and more of an effect in the oviraptorids (Fig. 3a–c) compared with *Incisivosaurus* (Fig. 3d) and the other theropods (Fig. 3e–g). The greatest difference between bilateral and unilateral bites occurs with the *Citipati* mid-palate bite position (Figs. 3c and 4).

The distribution of stress in the crania is broadly similar to the comparative bending test and differs chiefly with increased stresses in areas of jaw adductor muscle origination (Fig. 3). All oviraptorids show consistent stress hotspots across all bite positions in the supratemporal bar and squamosal, palatine, region of quadrate–pterygoid contact, and the anterior half of vomer (Fig. 3a–c). There are noticeable hotspots at the bite points on the premaxilla in the anterior and mid-palate scenarios; the main body of the premaxilla shows very little stress (Fig. 3a–c). The oviraptorids show some degree of stress in the processes of the premaxilla above the antorbital fenestrae (Fig. 3a–c). There is some stress consistently in the anteroventral parts of the braincase, near and including the epipterygoids, and in the thin parasphenoid rostrum of all oviraptorosaurians (Fig. 3a–d).

The nasal process of the premaxilla generally shows very little stress in the oviraptorids (Fig. 3a–c)—only a small degree in *Conchoraptor* under anterior and posterior biting (Fig. 3a). The parietals vary in stress among the oviraptorids; low in *Conchoraptor*, medium in *Khaan*, higher in *Citipati*, and consistent across bite point scenarios (Fig. 3a–c). The maxillae show

**Fig. 3 | Von mises stress distribution from biting test using estimated muscle force vectors.** Von Mises stress (MPa) contour plots from FEA modelling different bite positions in cranial models of oviraptorid oviraptorosaurians *Conchoraptor* (**a**), *Khaan* (**b**), *Citipati* (**c**), early diverging oviraptorosaurian *Incisivosaurus* (**d**), and non-oviraptorosaurian theropods *Allosaurus* (**e**), *Erlikosaurus* (**f**), and *Ornithomimus* (**g**). Force vectors were estimated from volumetric adductor muscle reconstruction and bite points modelled with bilateral constraints for three different locations on the palate and unilaterally at the mid-palate. Scale bars on the right are 50 mm.

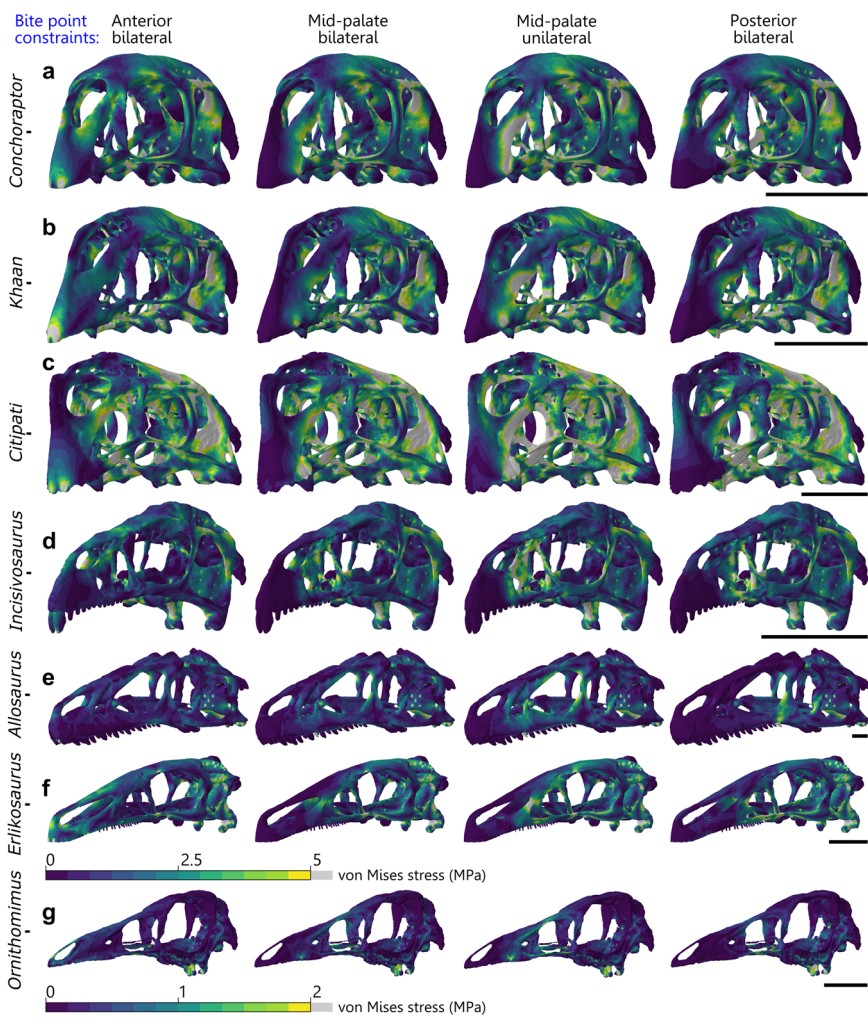

**Fig. 4 | Von Mises stress data from biting test using estimated muscle force vectors.** Mean values and whiskers showing first quartile, median, and third quartile of von Mises stress (MPa) from FEA using estimated adductor muscle forces modelling biting at three positions on the palate of cranial models of oviraptorosaurians *Conchoraptor, Khaan, Citipati,* and *Incisivosaurus,* along with *Allosaurus, Erlikosaurus,* and *Ornithomimus.* Ant = anterior, Mid = mid-palate, Pos = posterior.

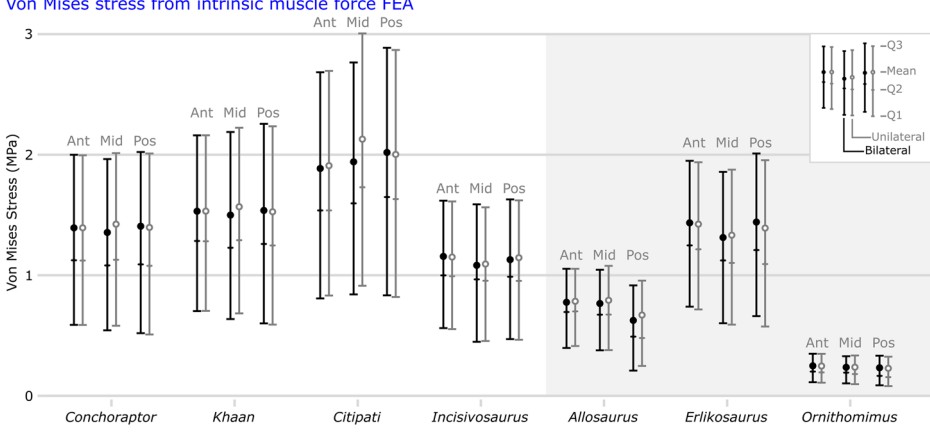

fairly high stress in all bite points for *Citipati* (Fig. 3c), unlike other oviraptorids (Fig. 3a, b). This is the area where stress is especially exaggerated by unilateral mid-palate bites in *Conchoraptor* (Fig. 3a) and especially *Citipati* (Fig. 3c), with essentially the entire region surrounding the antorbital fenestrae experiencing high stresses. The unilateral mid-palate bite is also the only scenario where an oviraptorid, *Citipati*, shows high stresses in the nasal (Fig. 3c) (and the contribution of the premaxilla to the nares).

Stress is noticeably lower in *Incisivosaurus* (Figs. 3d and 4) compared with the oviraptorids (Fig. 3a–c) but likewise occurs around the supratemporal bar and squamosal, and the region of the pterygoid–quadrate contact (Fig. 3d). The nasals and lacrimals show some stress in anterior biting, while in mid-palate and posterior biting stress is expressed more on the interfenestral bar of the maxilla, the lacrimal, and the postorbital bar (Fig. 3d). Unilateral biting chiefly exaggerates stresses in the interfenestral bar of maxilla and the lacrimal in the mid-palate and posterior scenarios (Fig. 3d and Supplementary Fig. S2), but only by a small amount. Unlike in the oviraptorids, the more robust jugal of *Incisivosaurus* consistently shows low stress (Fig. 3d). When constrained at the prominent

front teeth for an anterior bite, the teeth show fairly low stress, and the premaxillae show negligible stress in all bite scenarios (Fig. 3d).

## Discussion

The bending test using scaled loading on the oviraptorid cranium supports our first hypothesis that Oviraptoridae developed a relatively stronger cranial morphology than other herbivorous theropods. This is to a similar or greater relative degree as the cranium of the carnivorous *Allosaurus* (Fig. 2), which has been previously assessed as overengineered[58]. These findings quantify the strength of the oviraptorid cranial morphology and build on the FEA findings of Ma et al.[56]. Both the crania and mandibles[56] of oviraptorids display high stress resistance compared with other non-avialan theropods to facilitate high bite forces[17].

Oviraptorid crania achieve stress resistance through cranial shortening and stresses being borne by robust posterior processes of the premaxilla below the nares, rather than in the nasals and frontals dorsal to the antorbital fenestra. The contour plots indicate that stress is also distributed ventrally from the rostrum via the modified longitudinally directed connection of the pterygoid and ectopterygoid with the maxilla[71,72]. It is also distributed via the short stout vomer into the pterygoids which are robust (Figs. 1 and 3). The effect of this is that the most pneumatic areas of the cranium (nasal and frontal) and the reduced jugal–quadratojugal bar experience very low stress, and the cranium is a more efficient structure compared with other herbivorous theropods (lower total strain energy, less energy lost to bending).

Despite its peculiar intermediate morphology[44], the cranium of the early diverging oviraptorosaurian *Incisivosaurus* is characterised by stress patterns in response to bending that more closely resemble later diverging oviraptorids than other toothed theropod dinosaurs (Figs. 1 and 3) or any intermediate condition. This suggests that the cranium of *Incisivosaurus* already has several of the functional adaptations that characterise the response of the oviraptorid cranium to bending forces, and these were therefore established at the base of Oviraptorosauria.

In *Incisivosaurus*, enlargement of the orbit and the beginnings of a shortened skull and reinforced palate also funnel stresses away from the jugal–quadratojugal bar to other areas (Figs. 1 and 3). Low stresses suggest that the more massive, typical theropod-like jugal in *Incisivosaurus* is not a structural limiting factor in cranial function and is in a position to be adaptively reduced towards the more minimal morphology of the oviraptorids.

The development of heterodonty is characteristic of dinosaur taxa spanning a trophic shift and/or being likely omnivorous/herbivorous[30]. Furthermore, conical/incisiform teeth are associated with areas of subsequent tooth loss in several herbivorous lineages[30]. Specialised tooth types and arrangements such as the loss of pronounced replacement waves and gaps between teeth (a more continuous cutting surface) may function as analogous to a beak as plant material is adopted into the diet, a precursor to later evolution of a rhamphotheca[30]. This could be the case in *Incisivosaurus* with the elongate premaxillary teeth and more uniform close-set lanceolate maxillary teeth respectively functioning like tip and lateral cutting margins of a beak. This beak-like heterodonty combined with increased cranial stress resistance may have laid the foundation for modification towards the oviraptorid cranial morphology across a trophic shift.

The incisor-like front teeth of *Incisivosaurus* bear prominent wear facets on their lingual surface, interpreted by Xu et al.[44] as a possible indicator of herbivory. The smooth, rounded wear facets appear to be the product of grinding or wear from repetitive contact, rather than the more traumatic spalling that can occur on teeth from carnivorous forms from feeding[73]. However, it is unclear what these teeth were wearing against; the anterior tip of the *Incisivosaurus* mandible is edentulous and appears correctly positioned with the dentary tip posterior to the front cranial teeth for the maxillary and dentary tooth rows to occlude properly. It is possible the mandible terminated with a keratinous covering[44,45] but contact with softer keratin to produce wear facets on the harder enamel of the front teeth seems unlikely.

The comparatively good performance of the *Incisivosaurus* cranium under the head-twisting scenario (Supplementary Figs. S3 and S4) may suggest the elongate front teeth could be used robustly as a tool, creating these lingual wear facets against foodstuffs and the environment utilising neck musculature. The relatively long roots of the teeth may be related to this function—superficially similar long-rooted mandibular teeth in rodent mammals do not reduce stress distribution across the mandible but are driven by factors such as rapid incisor wear[74], though tooth replacement is a factor in *Incisivosaurus* and replacements for both front teeth are present within the premaxilla[45]. Nevertheless, the models here suggest pulling these front teeth against material with posteriorly directed head movement would create more stresses (chiefly in the teeth themselves and the palate and pterygoids) than a twisting motion of the same force (Supplementary Fig. S3), though less than that comparatively experienced by the crania of *Erlikosaurus* and *Ornithomimus*. In retaining small maxillary teeth, *Incisivosaurus* may have had more capacity than some of the oviraptorids for orally processing food. Experiencing the lowest stresses of any bite position when modelling a mid-palate bite (Fig. 4) on this tooth row further suggests that its cranium was adapted for biting/chewing in this way, either bilaterally or unilaterally.

*Citipati* appears to differ functionally from the other two oviraptorids tested. Its cranium appears relatively weaker in our comparative bending test, experiencing higher stresses than the other oviraptorids. It is sometimes even higher than the earlier diverging *Incisivosaurus*, especially under unilateral loading to the edge of the mid-palate (Fig. 2). The *Citipati* beak is relatively wider than the other oviraptorids, positioning unilateral force relatively further from the midline of the cranium. This increase in stress includes notably greater stresses in the otherwise minimally stressed nasal bones; the most stressed the delicate pneumatic nasals become in any of the scenarios for the oviraptorids (Figs. 1c and 3c).

In the muscle-driven test, *Citipati* shows increased stress in a posterior biting position relative to the other bite locations (Fig. 4). Though the relative difference is fairly small (approximately a 7% increase in mean stress from anterior to posterior), and the larger size of the *Citipati* cranium relative to the other oviraptorids could feasibly lead to a larger degree of relative difference, a similar pattern is absent when comparing between bite points for the other oviraptorids (Fig. 4 and Supplementary Table 3). This may suggest it was less adapted for crushing food with the posterior part of its palate and more adapted for anteriorly focussed biting or cropping with its wide blunt premaxilla—though the actual functional margin and occlusion of the *Citipati* jaws is uncertain as the rhamphotheca shape is unknown. Meade and Ma[17] suggested the *Citipati* adductor muscle architecture was potentially more adapted for a vertical bite with less emphasis on anteroposterior grinding jaw movement, which could support anteriorly cropping jaw function. The uniquely large discrepancy between the mid-palate bilateral and unilateral bites in *Citipati* indicates its cranial morphology was especially poorly adapted for uneven biting in this region and symmetrical application of bite force was likely important to its cranial function. This may indicate a need for controlled biting of static foodstuffs and suggest the capture of moving/struggling prey was particularly unlikely/uncommon in *Citipati*. This is further demonstrated by higher stresses under the extrinsic twisting scenario compared with the other oviraptorids (Supplementary Figs. S3 and S4).

*Erlikosaurus* was also found to be adapted to feeding at the jaw tip[60] but in a way that may have more harnessed neck musculature to compensate for a lack of jaw adductor muscle power. *Citipati* did not lack jaw adductor muscle power but has one of the best adapted cranial morphologies tested for the head-pulling scenario (Supplementary Figs. S3 and S4). Combined with its adaptations for a powerful front-beak bite, stripping or peeling of plant material is a possible functional interpretation. The other oviraptorids, *Khaan* and *Conchoraptor*, appear better suited towards more generalised cranial function and diet. They perform equally well at all bite points of their palate and experience very similar mean von Mises stress in all the bite and extrinsic scenarios (Figs. 1a, b and 2a, b).

It is important to caveat our FEA scenarios are inherent simplifications of anatomy and feeding dynamics, as is necessary[75]. The loading in our scenarios incorporating muscle force vectors assumes full simultaneous contraction of all jaw adductor muscles and identical isotropic and homogenous material properties for bone in each taxon. We do not model cranial sutures which can affect FEA results[76] such as relieving strain locally at the expense of elevated strain in other regions of the cranium—and the reverse[77]. Nevertheless, in testing our second hypothesis, our results show marginally greater cranial stresses in the oviraptorid oviraptorosaurians than in the other theropods when modelling biting using intrinsic muscle forces (Figs. 3 and 4). This suggests that, despite the increased relative strength of their cranial morphology, it is not enough to compensate entirely for the greatly increased relative force of their jaw adductor musculature[17].

Mammalian cranial bone is shown to operate at safety factors (ratio of the strength of biological component to the maximum load it is expected to withstand during life) ranging from 1.8 to 11[78], suggesting that the greater cranial stresses (much less than an order of magnitude; Fig. 4) modelled in the oviraptorids in comparison to other theropods are not outside the scale of feasible variation within biological groups. It may be that functional or developmental pressures towards an unusual pneumatic cranial shape in oviraptorids (for uncertain reasons), along with pressures to facilitate the musculature for powerful biting, combined to produce a cranial structure adapted to functioning slightly closer to structural safety limits compared with other theropods.

It is feasible that adaptation towards a light pneumatic cranium may be linked with the relatively long neck and cursorial body plan of later diverging oviraptorosaurians. A light cranium is coupled with a long flexible neck in modern ostriches providing mobility to accomplish varied tasks[79,80]. A lighter cranium may have reduced the need for a long counter-balancing tail (oviraptorid tails are short and stocky[81]), perhaps allowing for a more erect posture and assisting stability of the head while running at high speeds (similar to modern ostriches[80]). Elongate arctometatarsalian hindlimbs have been interpreted as an adaptation for cursoriality and speed for predator avoidance in avimimid[27] and caenagnathid oviraptorosaurians, and for pursuing small prey[27,82,83]; see Rhodes et al.[84], for alternate wading interpretation. Limbs may be proportionally shorter in oviraptorids[85], but have nevertheless been reconstructed as having robust caudofemoral musculature[81] and they were likely also fast-moving[1,81,86,87].

Adaptation towards strong bite forces is linked with the development of orbital shapes that provided extra cranial stress resistance in some carnivorous archosaurs[88] (see the elliptical shape of the *Allosaurus* orbit; Fig. 1e), but oviraptorids retain a large circular orbit relative to cranial size[19]. Circular orbits are common in herbivores but are a relatively weak orbital morphology[88]. Retention of a large circular orbit in oviraptorids despite adaptation towards powerful bites may suggest selective pressures related to visual range or acuity competing with factors related to diet in the evolution of their cranial function[89].

Reconstructing a keratinous covering on the beak of the oviraptorids reiterates the findings of other studies; a rhamphotheca reduces mechanical stresses directly under the areas it covers[42,60]. In *Erlikosaurus* and *Ornithomimus*, a keratinous rhamphotheca supports some of the most highly stressed areas of the cranium during anterior loading (see also refs. 40,42,60), but in *Conchoraptor*, *Citipati*, and *Khaan*, the regions presumably covered by a rhamphotheca do not broadly experience high stresses other than the immediate position of the bite point constraints themselves, especially compared to the posterior half of the cranium (Figs. 1 and 3). Stress reduction broadly in the premaxilla and narial region may, therefore, be less of a driving factor in the evolution of the rhamphotheca in Oviraptoridae compared with Therizinosauria and Ornithomimosauria. It may be the case that the development of a rhamphotheca in oviraptorosaurians was linked with an adaptive pressure towards having a feeding apparatus that was more continuously replaced in terms of wear (and potentially self-sharpening) to function more reliably and consistently. In this way, they would function in an analogous way to the beaks of modern parrots (though lacking their degree of cranial kinesis[90]), which can be entirely replaced in a few months[91].

Initial selection for tooth loss in oviraptorosaurians was likely not related to weight reduction (the effect is small[60,92]) and may have been a side effect of selection for fast embryo growth and thus shorter incubation time[93]. This would have been particularly advantageous for clades like oviraptorosaurians which are known to have brooded nests of eggs[8,94–96]. Reducing the time before embryo hatching would make this less energetically costly behaviour. Although influences of developmental biology and dietary ecology are not mutually exclusive, selection towards a toothless beak purely linked with a dietary shift or specialisation seems unlikely. The subsequent lack of reliance on the mechanical benefit from a keratinous rhamphotheca to cranial strength broadly in oviraptorids may leave its morphology more readily adapted by other factors. Indeed, the diversity of beak shapes in modern birds has been largely contingent on trade-offs and constraints[97,98] rather than dominated by dietary effects. Functional influences on the rhamphotheca in addition to those linked to diet in oviraptorosaurians may include roles as a possible sensory organ[99] or thermoregulatory organ[100,101] as seen in modern birds. This latter factor may apply particularly in oviraptorosaurians that possessed high vascularised crests (e.g., *Corythoraptor jacobsi*[22]; *Rinchenia mongoliensis*[1]), which may have functioned as structures to offload heat at high temperatures or restrict heat loss at lower temperatures.

Cranial kinesis like that in modern birds seems impossible in the crania of oviraptorids[102]. Nevertheless, several movable units were suggested by Barsbold[12] but seem highly unlikely. A more restricted moveable articulation between the quadrate and quadratojugal was suggested by Lü for *Heyuannia huangi*[103] and *Nemegtomaia barsboldi*[104,105]. This would require the quadrate to be mobile, which is contrary to interpretations that the otic capitulum is immovably fixed to the braincase wall and in tight extensive contact with the squamosal in other oviraptorids[72,102,106]. Furthermore, the quadratojugal appears in tight elongate contact rostrally with the jugal[18], and its contact with the quadrate is either fused or strengthened by a deep quadratojugal cotyla on the quadrate and large quadrate condyle on the quadratojugal, and thus likely immovable in oviraptorids[102]. Holliday and Witmer[107] interpret the quadrate of most non-avian Maniraptoriformes (including oviraptorosaurians) as slightly kinetically competent (with synovial basal and otic joints and protractor muscles), yet not kinetic because they lacked permissive kinematic linkages. Similar kinetic competency has been investigated using FEA in *Tyrannosaurus rex*, finding it functionally akinetic[108].

The results of this study may not apply to the other family of later diverging oviraptorosaurians, Caenagnathidae, for which cranial remains are poorly known. Their jaws are characterised more by a longer rostrum and lower mechanical advantage, interpreted as adaption towards jaw closing velocity for prey capture as part of an omnivorous diet with more carnivory than oviraptorids[15,109]. On this theme, the thin rod-like jugal of the oviraptorids was one of the more highly stressed regions of their cranium during both the biting FEA tests and the extrinsic head-shake scenario (which may replicate the capture and killing of struggling prey). The jugal is one of the few preserved cranial elements of the relatively large North American caenagnathid *Anzu wyliei* and has a more robust and conventional theropod shape[110] compared to those seen in Oviraptoridae[18,27]. This may hint at cranial morphologies existing among Caenagnathidae with plesiomorphic adaptations to deal with struggling prey as part of a more carnivorous diet.

## Data availability

The datasets, including cranial 3D models and files for finite element analysis, generated and analysed during the current study are available from the Zenodo data repository for download at: https://doi.org/10.5281/zenodo.10379298.

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

## Acknowledgements
We thank Khishigjav Tsogtbaatar (MPC), Jolanta Kobylinska (ZPAL), Vladimir Alifanov (PIN), and Xing Xu (IVPP) for access to specimens in their care. We thank Andrew Cuff for 3D model data, Waisum Ma for discussion and photogrammetric data, and Jordan Bestwick, Richard Butler, Philip Cox, and Sam Giles for discussion on this study. This work was supported by a NERC doctoral studentship to L.E.M. through the Central England NERC Training Alliance (CENTA; Grant reference NE/L002493/1). M.P.'s participation in this study was supported by the School of Life Sciences of The Chinese University of Hong Kong.

## Author contributions
L.E.M., M.P. and S.L. conceived and designed the work; S.L. and A.B. contributed digital models and data; L.E.M. created the oviraptorosaurian 3D models, performed the finite element analyses, and analysed the data; L.E.M., M.P., A.B. and S.L. wrote the paper. All authors gave final approval for publication.

## Competing interests
The authors declare no competing interests.
