## [Peer review file · Communications Biology]

Reviewers' comments:

Reviewer #1 (Remarks to the Author):

This paper performs finite-element analyses of the crania of various oviraptorosaurian taxa and compares them to other theropods, in order to investigate the functional performance of the distinctive oviraptorid cranium. Overall, I think this is a very good submission as-is - the experiments were all performed and reported adequately, and the results interpreted appropriately. The main result - that the crania of oviraptorids are more similar in functional performance to 'typical' carnivorous theropods than to other herbivorous taxa is further very interesting, suggesting surprising functional conservatism despite their unusual morphology. This is interesting both in the context of theropod evolution, but also more broadly as a general example of the pitfalls in inferring function from form, and in understanding what mechanisms may underpin morphological variation. These results and their significance are well-explained (the discussion of tradeoffs in particular is very good), the methods section is more than sufficiently detailed to allow replication, and the figures are appropriate and helpful.

I am then broadly happy with the paper in its current form, pending one very minor comment regarding the discussion (see below). Consequently, I am happy to recommend it for publication pending possibly very minor revisions (see below) - a very good submission!

Specific comments:

Lines 371-384: How sure are you that the minor differences in performance between Citipati and the other taxa are actually ecologically significant? They are very similar to the other taxa, and higher peak/mean forces probably are not surprising just given that Citipati is notably larger than the other two? I know this is an issue that pervades all palaeontological biomechanical studies, and it certainly is not a big issue that harms the overall paper, but I cannot help but think that - especially given the other results of the paper - these results may indicate non-functional explanations for the differences in skull shape among these taxa?

Reviewer #2 (Remarks to the Author):

The work submitted by Meade and colleagues presents results on cranial functional morphology in oviraptorosaurs. The work takes advantage of previous works of the authors (particularly first author) about this dinosaur group.

My main concern on this manuscript is that authors use solid cranial models, and don't take into consideration sutures and/or skull kinesis (sutures are not mentioned on the whole manuscript and kinesis is not considered). For some years it has been ignored, sometimes based that requires very powerful computers, but nowadays it has no sense to ignore them, as demonstrated by several works on different reptiles clades, including dinosaurs (e.g. Moazen et al. 2009; Curtis et al. 2013; Jones et al. 2017; Wilken et al. 2019; Cost et al. 2020).

The inclusion of sutures (or functional units based on sutures) demonstrated that results are very different if sutures are considered. For that reason, the results herein presented - based on comparisons between models surely would be completely different if sutures and kinesis are considered. As results would be different, discussion and implications could be very different. For that reason, I strongly suggest authors to consider sutures (or functional units based on sutures) prior to publish this work.

In my opinion it's surprising that authors compare oviraptorosaurs with *Allosaurus*, not considering in any case cranial sutures: *Allosaurus* has been one of the first dinosaurs to be investigated based on its cranial suture morphology, demonstrating how sutures affect stress distribution (Rayfield, 2005).

I'm not oviraptorosaurian expert, but authors state at some points the similarities between oviraptorosaurs and birds (parrots in particular) and birds present kinetic skulls. Recently it was published a work on *T. rex* considering that skull was kinetic - and comparing it with birds (Cost et

al. 2020), so I assume that oviraptors would be also kinetic. Are oviraptors skulls kinetic? It would be nice to answer this question, and if these animals had kinetic skulls, to be considered on the analyses.

Another minor concern is about the models: some of them are derived from CT scanners while others come from photogrammetrical ones. It has been shown that results are affected based on the origin of the models (Rowe and Rayfield, 2022). Did authors validate in any manner if these affected their results?

References

- Rayfield (2005): <https://doi.org/10.1002/ar.a.20168>
Moazen et al (2009): <https://doi.org/10.1098/rspb.2008.0863>
Curtis et al (2013): <https://doi.org/10.1098/rsif.2013.0442>
Jones et al (2017): <https://doi.org/10.1098/rsif.2017.0637>
Wilken et al (2019): <https://doi.org/10.1242/jeb.201459>
Cost et al (2020): <https://doi.org/10.1002/ar.24219>
Rowe and Rayfield (2022): <https://doi.org/10.7717/peerj.13760>

Reviewers' comments:

Reviewer #1 (Remarks to the Author):

This paper performs finite-element analyses of the crania of various oviraptorosaurian taxa and compares them to other theropods, in order to investigate the functional performance of the distinctive oviraptorid cranium. Overall, I think this is a very good submission as-is - the experiments were all performed and reported adequately, and the results interpreted appropriately. The main result - that the crania of oviraptorids are more similar in functional performance to 'typical' carnivorous theropods than to other herbivorous taxa is further very interesting, suggesting surprising functional conservatism despite their unusual morphology. This is interesting both in the context of theropod evolution, but also more broadly as a general example of the pitfalls in inferring function from form, and in understanding what mechanisms may underpin morphological variation. These results and their significance are well-explained (the discussion of tradeoffs in particular is very good), the methods section is more than sufficiently detailed to allow replication, and the figures are appropriate and helpful.

I am then broadly happy with the paper in its current form, pending one very minor comment regarding the discussion (see below). Consequently, I am happy to recommend it for publication pending possibly very minor revisions (see below) - a very good submission!

Specific comments:

Lines 371-384: How sure are you that the minor differences in performance between Citipati and the other taxa are actually ecologically significant? They are very similar to the other taxa, and higher peak/mean forces probably are not surprising just given that Citipati is notable larger than the other two? I know this is an issue that pervades all palaeontological biomechanical studies, and it certainly is not a big issue that harms the overall paper, but I cannot help but think that - especially given the other results of the paper - these results may indicate non-functional explanations for the differences in skull shape among these taxa?

We thank Reviewer #1 for taking the time to read our manuscript and their positive remarks.

Our discussion of differing performance in this section is based on comparing between stress values for different bite points within each taxon, rather than comparing the stress values of a certain bite point between taxa. It should therefore be relative and not necessarily impacted by the absolute size of the cranium. The final sentence of this passage that relates to supplemental extrinsic FEA results directly comparing between taxa uses scaled forces, and we state in the methods for these that this removed the effect of size in comparison.

We do think it is important to acknowledge the differences in mean stress between bite points that we base this discussion on are small, however, and potentially the relative degree of difference between different bite points could indeed be a factor of cranial size when comparing between taxa.

We have adjusted **Lines 379-384** to better state this.

Reviewer #2 (Remarks to the Author):

Response to reviewers

The work submitted by Meade and colleagues presents results on cranial functional morphology in oviraptorosaurs. The work take advantage of previous works of the authors (particularly first author) about this dinosaur group.

My main concern on this manuscript is that authors use solid cranial models, and don't take into consideration sutures and/or skull kinesis (sutures are not mentioned on the whole manuscript and kinesis is not considered). For some years it has been ignored, sometimes based that requires very powerful computers, but nowadays it has no sense to ignore them, as demonstrated by several works on different reptiles clades, including dinosaurs (e.g. Moazen et al. 2009; Curtis et al. 2013; Jones et al. 2017; Wilken et al. 2019; Cost et al. 2020).

The inclusion of sutures (or functional units based on sutures) demonstrated that results are very different if sutures are considered. For that reason, the results herein presented - based on comparisons between models surely would be completely different if sutures and kinesis are considered. As results would be different, discussion and implications could be very different. For that reason, I strongly suggest authors to consider sutures (or functional units based on sutures) prior to publish this work.

In my opinion it's surprising that authors compare oviraptorosaurs with Allosaurus, not considering in any case cranial sutures: Allosaurus has been one of the first dinosaurs to be investigated based on its cranial suture morphology, demonstrating how sutures affect stress distribution (Rayfield, 2005).

*I'm not oviraptorosaurian expert, but authors state at some points the similarities between oviraptorosaurs and birds (parrots in particular) and birds present kinetic skulls. Recently it was published a work on *T. rex* considering that skull was kinetic - and comparing it with birds (Cost et al. 2020), so I assume that oviraptorosaurs would be also kinetic. Are oviraptorosaurs skulls kinetic? It would be nice to answer this question, and if these animals had kinetic skulls, to be considered on the analyses.*

We agree with Reviewer #2 that sutural connections can be an important aspect of assessing hypotheses of cranial functional morphology when modelled in FEA, particularly when concerning cranial kinesis.

Our decision to not incorporate sutures into our FE models was not one of computer power. Unfortunately, the position and anatomy of many of the sutural connections in the oviraptorosaurian crania were unclear in the CT datasets that form the basis of this work, either being obscured by still matrix-bound areas with low CT contrast or so damaged as to be unrecognisable in large regions (e.g. the entire cranial roof of *Khaan*) with only the general morphology of the bony elements able to be reconstructed.

We did not feel confident enough in reconstructing a significant amount of the cranial sutural connections consistently between all taxa to warrant including them in our analyses. Nor did we intend here to test a specific hypothesis of cranial kinesis, as in Cost et al. (2020), which could have utilised a more limited set of anatomical areas or functional units connected with modelled sutures.

This is in addition to that the fact that the comparative dataset of FE models we sourced were from previously published FEA studies that also did not include cranial sutures (*Allosaurus*, Montefeltro et al., 2020; *Erlikosaurus*, Lautenschlager et al., 2016; *Orthithomimus*; Bestwick et al., 2022). Significant extra study using the original CT datasets

Response to reviewers

that these models were derived from would have been necessary to accurately incorporate sutures into these comparative models.

Ultimately, we saw more value in performing a comparison between more crania modelled solidly, rather than a fewer number of crania modelled incorporating sutures which may indeed have lacked accuracy. We now detail this in our methods section (Lines 102-107) and add a caveat to our discussion section (Lines 408-414).

On the topic of cranial kinesis in relation to birds and in the context of previous work done on *Tyrannosaurus* (Cost et al., 2020), we reject the idea of substantial cranial kinesis in any sense like modern birds in oviraptorosaurians. It is important to note the conclusions of Cost et al. (2020) found *Tyrannosaurus* to be functionally akinetic. We now take this opportunity to briefly review the topic of cranial kinesis in oviraptorids in our discussion (Lines 484-498) suggesting any degree of kinesis in oviraptorosaurian crania would be negligible and, if present, likely be of a similar mode to the other non-oviraptorosaurian theropods among Maniraptoriformes. We already state cranial similarity of oviraptorids to birds is superficial (Line 38) and now reiterate the lack of parrot-like kinesis in relevant section (Line 463).

- Lautenschlager S., Brassey C.A., Button D.J., Barrett P.M. 2016 Decoupled form and function in disparate herbivorous dinosaur clades. *Scientific Reports* 6(1), 26495.

-Cost, I.N., Middleton, K.M., Sellers, K.C., Echols, M.S., Witmer, L.M., Davis, J.L. and Holliday, C.M., 2020. Palatal biomechanics and its significance for cranial kinesis in *Tyrannosaurus rex*. *The Anatomical Record*, 303(4), pp.999-1017.

-Montefeltro F.C., Lautenschlager S., Godoy P.L., Ferreira G.S., Butler R.J. 2020 A unique predator in a unique ecosystem: modelling the apex predator within a Late Cretaceous crocodyliform-dominated fauna from Brazil. *Journal of Anatomy* 237(2), 323-333.

-Bestwick J., Jones A.S., Nesbitt S.J., Lautenschlager S., Rayfield E.J., Cuff A.R., Button D.J., Barrett P.M., Porro L.B., Butler R.J. 2022 Cranial functional morphology of the pseudosuchian *Effigia* and implications for its ecological role in the Triassic. *The Anatomical Record* 305(10), 2435-2462.

Another minor concern is about the models: some of them are derived from CT scanners while others come from photogrammetrical ones. It has been shown that results are affected based on the origin of the models (Rowe and Rayfield, 2022). Did authors validate in any manner if these affected their results?

This was an important consideration when creating our retrodeformed cranial models. The difference in FEA results between photogrammetric and CT derived models comes from the fact photogrammetric models only capture surface geometry – they exclude internal structures or voids that may be structurally significant in FEA. The use of photogrammetric data was only necessary in the case of one taxon, *Conchoraptor*, as the premaxilla, maxilla, and majority of the palate are missing from the CT scanned dataset we had of this species. Fortunately, these areas lack pneumatization or other significant internal structures that may affect FEA results; thus, we were confident to incorporate photogrammetric versions into our model. The other oviraptorid models can also be seen from CT data to lack relevant internal anatomy in these areas. The areas with the most significant internal anatomy in *Conchoraptor*, those that are heavily pneumatized and the endocranial space of the braincase, are all derived from CT data.

We have made minor edits to Lines 97-101 touching on this point. A much more detailed methodology of the retrodeformation and creation of the cranial 3D models is given

Response to reviewers

previously in Meade and Ma (2022) and Meade (2023) and cited also in this location in our manuscript. We do not want to replicate aspects of these documents in the methods or supplement of this paper as they are extensive, and the methodology remains unchanged from these works.

-Meade, L.E. and Ma, W., 2022. Cranial muscle reconstructions quantify adaptation for high bite forces in Oviraptorosauria. *Scientific Reports*, 12(1), p.3010.

- Meade, L.E., 2023. Functional morphology of the oviraptorosaurian cranium (Doctoral dissertation, University of Birmingham).

REVIEWERS' COMMENTS:

Reviewer #2 (Remarks to the Author):

I acknowledge authors reply to my comments as they mostly give answers to all my questions. On one hand, the authors made a good job justifying on the rebuttal letter but also on the main text that, in their opinion, oviraptorosaurian skull is akinetic and for that reason it has no sense to consider this issue on their analyses.

On the other hand, I understand their decision to not analyze sutural connections as they are not preserved on the analyzed skulls (and I acknowledge they added this information on the main text), but it also implies that this is an important weak point of their analysis and consequently the quality of the results is lower than if sutures would be present and analyzed. However, I don't agree with them that "we saw more value in performing a comparison between more crania modelled solidly, rather than a fewer number of crania modelled incorporating sutures which may indeed have lacked accuracy" as the rest of models are complementary but not the principal group to be tested, the oviraptorosaurians. In other words, if sutures would be present, ideally could be analyzed with and without sutures, and in case results would be similar, then would not be a problem to also use the models without sutures. At the end, authors seem to accept that if sutures would be considered then results would be different and interpretation would be also different.

Overall, based on the information provided by the authors on the revised version the work is closed enough, and I don't suggest any other change as the analyses cannot be improved based on the specimens used. Hope in near future analyses could be performed in better preserved ones and results and interpretation can be assessed with confidence.